# Observation of second sound in graphite over 200 K

Zhiwei Ding [1,3], Ke Chen[1,3], Bai Song [1], Jungwoo Shin[1], Alexei A. Maznev[2], Keith A. Nelson [2✉] & Gang Chen [1✉]

Second sound refers to the phenomenon of heat propagation as temperature waves in the phonon hydrodynamic transport regime. We directly observe second sound in graphite at temperatures of over 200 K using a sub-picosecond transient grating technique. The experimentally determined dispersion relation of the thermal-wave velocity increases with decreasing grating period, consistent with first-principles-based solution of the Peierls-Boltzmann transport equation. Through simulation, we reveal this increase as a result of thermal zero sound—the thermal waves due to ballistic phonons. Our experimental findings are well explained with the interplay among three groups of phonons: ballistic, diffusive, and hydrodynamic phonons. Our ab initio calculations further predict a large isotope effect on the properties of thermal waves and the existence of second sound at room temperature in isotopically pure graphite.

[1] Department of Mechanical Engineering, Massachusetts Institute of Technology, Cambridge, MA 02139, USA. [2] Department of Chemistry, Massachusetts Institute of Technology, Cambridge, MA 02139, USA. [3]These authors contributed equally: Zhiwei Ding, Ke Chen. ✉email: kanelson@mit.edu; gchen2@mit.edu

Thermal transport in dielectrics and semiconductors is often mediated by the random-walk behavior of phonons and follows Fourier's law of heat diffusion. However, Fourier's law breaks down at sufficiently small length scales or low temperatures, where unusual regimes of ballistic and hydrodynamic phonon transport are observed. In the ballistic regime, phonons can travel a distance longer than the conduction length scale without scattering, leading to an effective thermal conductivity value that diminishes as the length scale is reduced. This regime has been studied extensively owing to its importance in applications such as electronics thermal management and thermoelectric energy conversion[1–4]. The hydrodynamic regime takes place when momentum-conserving normal scattering (N-scattering) is stronger than momentum-destroying resistive scattering (R-scattering), leading to a collective drift motion of phonons under a temperature gradient. This strongly correlated collective motion of phonons leads to peculiar thermal transport phenomena such as second sound[5–11], phonon Poiseuille flow[6,12,13] and Knudsen minimum[12,14], parallel to the hydrodynamic regime of strongly correlated electrons[15–18]. For over half a century, phonon hydrodynamic transport was deemed exotic and mattered only at extremely low temperatures. However, phonon hydrodynamics at substantially higher temperatures in low-dimensional and van der Waals materials has recently been theoretically predicted and experimentally observed, stimulating renewed interest, especially in second sound[12,13,19–25].

Second sound is the wavelike propagation of heat[26]. First observed by Peshkov in 1944 in superfluid $^3$He at 1.4–1.6 K[27] and explained with Landau's two-fluid model[28], it was later predicted to exist also in solids when phonon-phonon N-scattering dominates over R-scattering[8]. Experimentally, second sound in solids was initially observed via heat-pulse methods near liquid-helium temperatures, for example, at 1.2–4.0 K in Bi[29] and 10–18 K in NaF[30]. Some of us recently reported observation of second sound above 100 K in graphite using the transient thermal grating (TTG) technique[21] in which the thermal transport length scale is imposed by the use of crossed excitation laser pulses that produce an optical interference pattern of alternating peaks and nulls at the sample, resulting in a sinusoidally varying temperature "grating" profile. The observation of second sound in graphite is further confirmed via the transient hydrodynamic lattice cooling by picosecond laser irradiation of graphite[31]. The kinetics of transport from the heated transient grating peaks to the unheated nulls are measured through time-resolved diffraction of probe laser light from the surface modulation caused by thermal expansion at the heated grating peaks. Transport by second sound was expected to be observable at even higher temperatures for TTG spatial periods smaller than those used in our previous study as suggested by the theoretical calculations[21,32]. However, the temporal resolution of our continuous-wave-laser-probed TTG system (~0.5 ns) imposed a lower limit of the grating period of about 6 μm, below which the kinetics could not be resolved.

Theoretically, phonon-mediated second sound is described as damped temperature waves derived either from the phonon Peierls-Boltzmann transport equation (PBTE)[5,6,9,10] or equilibrium correlation functions[11,33]. In particular, the dispersion of second sound has been analyzed by applying a temperature perturbation to the PBTE[9] and analytically solving it for the quasi-momentum auto-correlation function[11]. However, all these efforts focus on the frequency windows within which we can observe second sound, while the wavevector dependence of the second-sound velocity, i.e., the dispersion of the velocity, has not been discussed.

While second sound originates from phonons in local thermal equilibrium caused by strong normal scattering (a drifting equilibrium)[6], ballistic phonons can also generate a wavelike heat flow, as was evident in past experiments on NaF and Bi[29,30], in which heat pulses due to ballistic longitudinal and transverse phonons reached the detectors before the arrival of second sound. These thermally excited phonons were sometimes mistakenly attributed to first sound, but it was shown[34] that heat-pulse excitation used in the earlier experiments cannot excite a mechanical wave of large amplitude. Mechanically excited first sound can be damped by phonons in local equilibrium via the Landau-Rumer[35] and Akhiezer mechanisms[36], because the wavelength of the first sound is much longer than the phonon thermalization length, i.e., the phonon mean free path (MFP). If the MFP is longer than the wavelength of the first sound, it was called zero sound, in analogy to the propagation of zero-sound waves across a Fermi liquid in the ballistic limit[37–40]. In analogy to the zero sound for mechanical sound waves, we call the ballistic thermal wave "thermal zero sound." In heat-pulse experiments[29,30], phonons are excited in all directions by the heater, but only those propagating within a small solid angle subtended by the detector are recorded, leading to thermal zero-sound velocities close to those of the longitudinal and transverse phonons. Oscillations due to thermal zero sound were also calculated theoretically by solving the PBTE under the constant MFP approximation for the TTG experimental geometry[41].

In this work, we report the observation of second sound at record-high temperatures of over 200 K via pulsed-laser-probed TTG measurements at grating periods of around 2 μm. The thermal transport that is measured also includes contributions from thermal zero sound due to ballistic phonons. We show that the transport can be viewed as a mixture of three groups of phonons: hydrodynamic phonons experiencing strong normal scattering, contributing to second sound; ballistic phonons contributing to thermal zero sound; and resistive phonons contributing to diffusion along the temperature gradient. Using exact solutions of the PBTE, we demonstrate increasing contributions of thermal zero sound to the TTG signal with decreasing grating period. We also predict a large isotope effect on second sound. Notably, room-temperature second sound is expected in isotopically enriched graphite. What is reported here is the "drifting" second sound as discussed by Hardy[5] where collective drift motion of phonons due to strong N-scattering is essential. This "drifting" second sound is distinguished from "driftless" second sound suggested by a recent report on measurements in a rapidly varying temperature field[42].

## Results

**Observation of second sound in graphite above 200 K.** As discussed in Ref. [21], second sound has recently been observed in graphite at around 100 K via TTG measurement with a continuous-wave (CW) probe laser. However, with a CW probe and the measurement bandwidth of Ref. [21], the time resolution was limited to ~0.5 ns which prevented measurements at smaller grating period where second sound is expected at much higher temperatures. In this work, we replaced the CW laser with a femtosecond pulsed laser and used the standard ultrafast measurement technique in which the pump-probe delay time is varied to acquire time-dependent signals[43,44]. The experimental details can be found in Material and methods and the Supplementary Materials (SM1). Following the Green function approach developed by Chiloyan et al.[45], we also theoretically simulate the TTG response using first-principle calculations with no fitting parameters (SM2).

As demonstrated in the simulated TTG signal (Fig. 1a), second sound is characterized by a sign reversal in the heterodyned TTG signal[21]. Shortly after TTG excitation, the diffracted signal field from the modulated surface profile in which thermal expansion

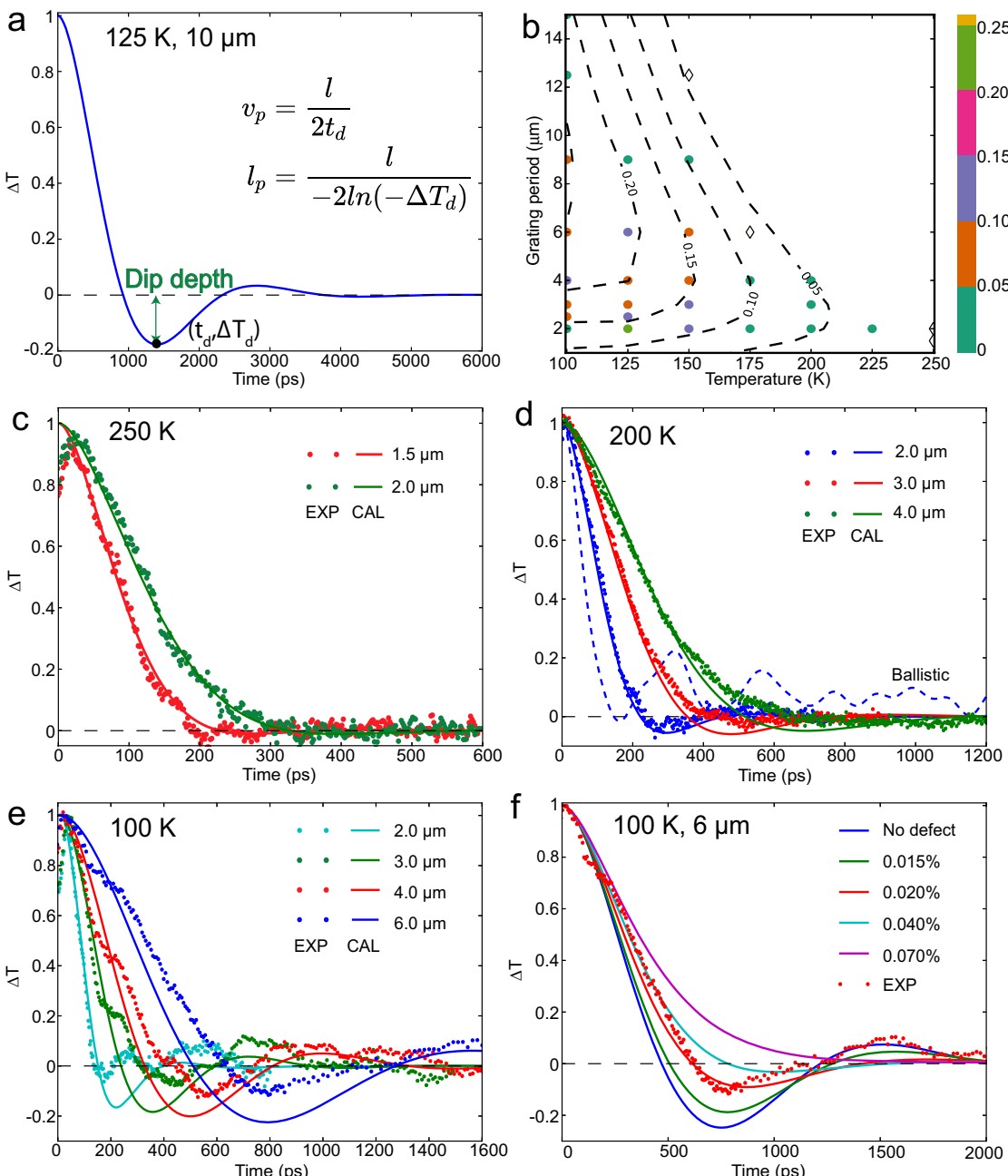

**Fig. 1 Measured (points) and simulated (curves) heterodyned TTG signals for graphite after normalization with respect to the peak height at early times. a** The simulated temperature response at 125 K with a 10 μm grating period. The negative dip is the hallmark of wavelike thermal transport and its depth represents the second-sound strength. **b** TTG second-sound temperature window at different grating periods. The color bar shows the normalized dip depth. The circles (diamonds) indicate observation (no observation) of wavelike transport. TTG signals at (**c**) 250 K, (**d**) 200 K, and (**e**) 100 K for various grating periods. **f** The calculated effect of vacancies on TTG signal at 100 K for a 6 μm period.

has occurred at the grating peaks is superposed constructively with a reference field, yielding a positive heterodyned signal. Second sound moves heat away from the peaks and into the nulls in wavelike fashion, not merely equalizing the temperature as in ordinary thermal transport but raising the temperature at the nulls above that of the peaks. Thermal expansion becomes more pronounced at the nulls than at the peaks, reversing the surface modulation profile and therefore reversing the phase of the diffracted signal field. The resulting destructive superposition with the reference field yields a negative heterodyned TTG signal, a clear signature of wavelike transport. The strength of the wavelike feature can be measured expediently by the ratio of the

negative dip depth to the initial signal peak height. Figure 1b summarizes experimental results recorded at different grating periods and temperatures, using the normalized dip depth as the metric for the wavelike contribution to transport. The calculated temperature windows for second sound at each grating period are also indicated. A normalized depth of about 0.05 marks the experimental limit below which we cannot clearly observe second sound. TTG signals often contain oscillations due to acoustic vibrations, which have much lower damping rates. We found that the acoustic signal could be suppressed by maximizing the peak height through adjustment of the reference field phase as discussed in SM1. At 250 K (Fig. 1c), any sign flip in the signal

is obscured by the acoustic signal at the experimental signal-to-noise ratio. But the clear hallmark of second sound is observed at 200 K (Fig. 1d) and 225 K (Fig. S3a). The calculated temperature windows (Fig. 1b) indicate that second sound can persist for longer thermal transport length scales at lower temperatures, which is consistent with the present and previous[21] experimental observations at 100 K (Fig. 1e). A larger discrepancy between experiment and simulation is observed at 100 K (Fig. 1e) than 200 K (Fig. 1d), which might result from the increasing importance of phonon-impurity scattering at lower temperatures compared to phonon-phonon scattering. At 100 K (Fig. 1f), about 0.02% vacancy density could explain the discrepancy between simulation and experiment, while the simulated response at 200 K showed a much smaller effect at the same level of defect (Fig. S4).

**Dispersion of measured thermal wave**. Based on the TTG signal, the velocity $v_p$ and propagation length $l_p$ of the thermal waves can be estimated from the dip position $t_d$ and normalized depth $\Delta T_d$ as (Fig. 1a):

$$v_p = \frac{l}{2t_d} \tag{1a}$$

$$l_p = \frac{l}{-2\ln(-\Delta T_d)} \tag{1b}$$

where $l$ is the grating period. Equation (1a) follows from the definition of the dip position as the time duration it takes for the thermal wave to travel from a high-temperature region to an adjacent low-temperature region, that is, half the grating period; Eq. (1b) arises from the fact that the amplitude of an exponentially decaying wave drops to $\exp(-l/2l_p)$ over a distance of $l/2$. Using Eq. (1), we determined the thermal-wave velocity and propagation length at different temperatures and grating periods, from both experimental and simulated TTG signals. As shown in Fig. 2a, b, there is a qualitative agreement between the simulations and experiment. The quantitative discrepancy is attributed to defects in our graphite samples.

Refs. [5–8] used Callaway model[46] to study second sound. The Callaway model splits the scattering term in the PBTE into two different relaxation terms: resistive scattering and normal scattering that drive the phonon distribution to the local equilibrium Bose-Einstein distribution and displaced Bose-Einstein distribution respectively. From the energy and momentum conservation equations, the second-sound velocity $v_{ss}$ and propagation length $l_{ss}$ in the x-direction can be written as[47]:

$$v_{ss}^2 = \frac{\left\langle \frac{C_q q_x v_x}{\omega_q} \right\rangle^2}{\left\langle \frac{C_q q_x^2}{\omega_q^2} \right\rangle \langle C_q \rangle} \tag{2a}$$

$$l_{ss} = \frac{2v_{ss} \left\langle \frac{C_q q_x^2}{\omega_q^2} \right\rangle}{\left\langle \frac{C_q q_x^2}{\omega_q^2 \tau_{qR}} \right\rangle} \tag{2b}$$

Where $C_q$, $q_x$, $v_x$, $\omega_q$, and $\tau_{qR}^{-1}$ are the mode-specific heat capacity, wavevector, group velocity, frequency, and R-scattering rate, respectively, and <> means summation over all the phonon modes i.e., $\left\langle C_q \right\rangle = \sum_q C_q$. We refer to the second-sound velocity and propagation length obtained from Eq. (2) as the intrinsic limits (Fig. 2a, b, d). To further understand the dispersion of the measured wave velocity, we also simulated the TTG temperature response at the ballistic limit, where phonons propagate without interactions. Wavelike behavior due to thermal zero sound is also seen in the calculated response in the investigated temperature range in the ballistic regime (Fig. 2c and Fig. S5), although the

wave contains multiple frequencies from different polarizations. A similar wave propagation velocity could be defined based on the position of the first dip and is referred to as thermal zero sound velocity. The thermal zero sound is faster than the intrinsic second-sound speed at all the investigated temperatures (Fig. 2d) as expected, because second sound involves mixing and scattering of different phonons to approach a local thermal equilibrium[5,6]. We can also consider longitudinal and transverse waves separately as these waves do not interact in the ballistic limit, as plotted in Fig. 2c. At 100 K, the flexural mode dominates the signal and its superposition with longitudinal and transverse waves leads to higher thermal zero sound velocity. Both the second sound velocity obtained from Eq. (2a) and the thermal zero sound velocity increase from 100 to 125 K first, then start to decrease upon further rise in temperature (Fig. 2d). However, the group velocity of the measured thermal wave, which includes both second sound and thermal zero sound, decreases monotonically with increasing temperature at a fixed grating period (Fig. 2a and Fig. S6a). As the grating period decreases, the measured wave propagation speed increases from the intrinsic second sound velocity to the thermal zero sound velocity (Fig. 2a) since a greater portion of the phonon MFPs exceed the grating spacing. On the other hand, the measured wave propagation length increases at longer grating periods (Fig. 2b) because of the reduced contributions of ballistic phonons with effective propagation lengths limited by the grating period.

The measured grating-period-dependent wave propagation speed yields a convex dispersion relation, which is consistent with the calculated dispersion (Fig. 2e, also Material and methods and SM4). In addition, the gap between the real and imaginary parts of the frequency is maximized at mid-wavevector, which is consistent with the trend of the second-sound strength metric (i.e., normalized dip depth) we defined (Fig. 2f).

**Three groups of phonons**. Phonons participating in thermal transport are a mixture of those experiencing no scattering, those experiencing both N- and R-scattering, and those experiencing R-scattering only[48]. The observed thermal wave is formed by a superposition of ballistic, diffusive, and hydrodynamic phonons. The dispersion and temperature-dependence of the measured thermal waves can thus be understood via the interplay among these three groups of phonons. Such a picture could also explain the heat-pulse experimental observations as detailed in SM5. To qualitatively describe how much the ballistic/diffusive component is contributing to the measured thermal waves, we estimate the fraction of the initially excited phonons with traveling distance shorter/longer than the corresponding MFP as:

$$f_b = \frac{<C_q v_x^2 f_b^q(\Lambda_o/d)>}{<C_q v_x^2>} \tag{4a}$$

$$f_d = \frac{<C_q v_x^2 f_d^q(\Lambda_R/d)>}{<C_q v_x^2>} \tag{4b}$$

where $d$ is the traveling distance, $\Lambda_o$ is the total MFP, considering all scattering processes, and $\Lambda_R$ is MFP considering only resistive scattering processes. Here $f_{b/d}$ is the fractions of ballistic and diffusive contribution to the transport. The fraction of hydrodynamic contribution $f_h$ could be approximated as: $1 - f_b - f_d$. $f_{b/d}^q$ is the ballistic/diffusive contribution function for phonon mode $q$. Details on the definition of traveling distance $d$ and function $f_{b/d}^q$ are provided in SM6. Figure 3 shows the approximated fractions of ballistic, hydrodynamic, and diffusive phonons defined at different grating periods and temperatures.

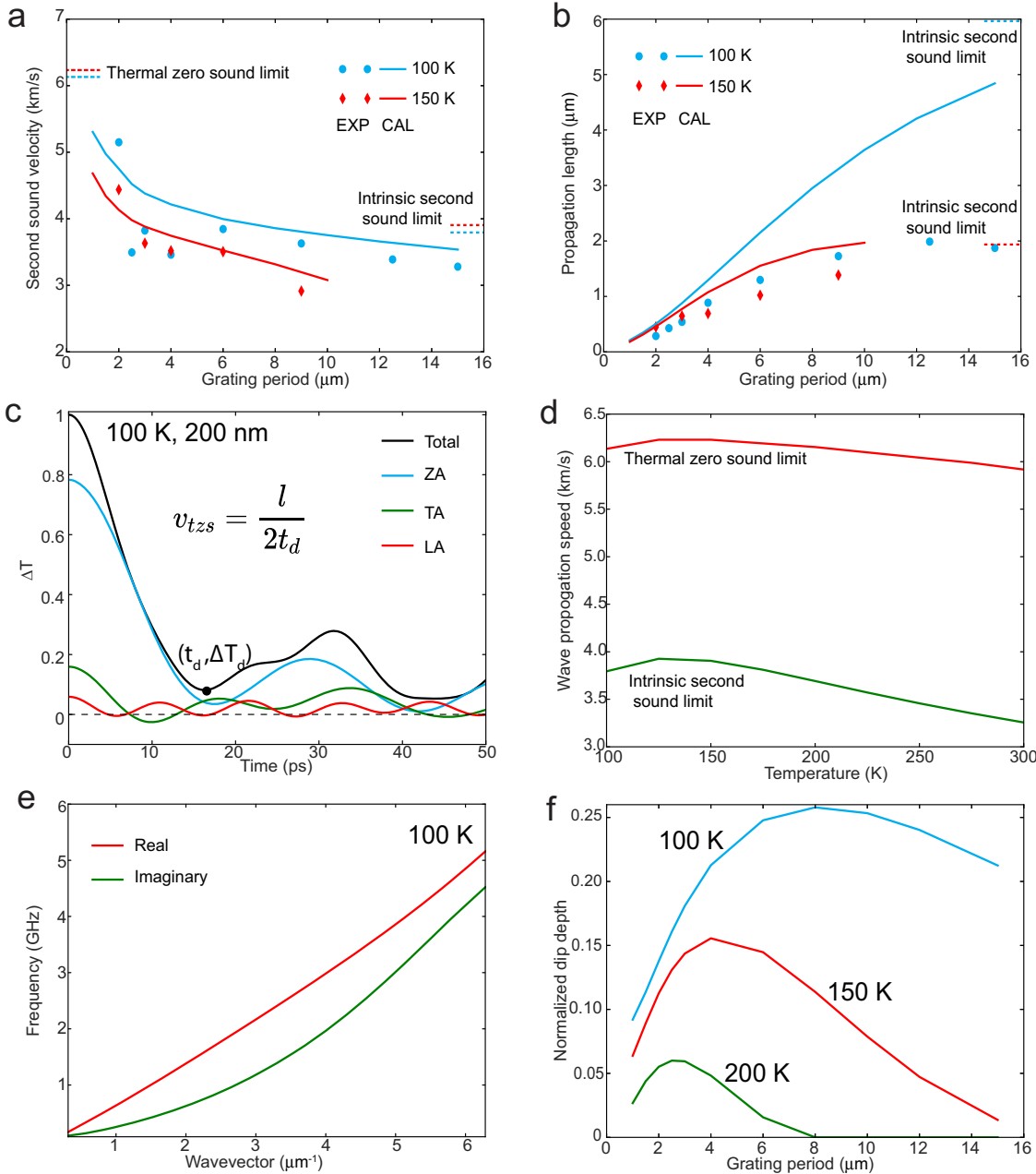

**Fig. 2 Dispersion of thermal waves.** The measured and calculated data are represented by points and curves, respectively. **a** Dependence of wave velocity on grating period. Also shown are the intrinsic second sound limit from Eq. (2) and the thermal zero sound limit discussed in Methods. Damping reduces the second sound velocity below the intrinsic limit at large grating periods, while zero sound increases the measured wave velocity at small grating periods. **b** Propagation length variation with grating period at different temperatures. **c** Calculated ballistic limit of the TTG signal at 100 K and its contributions from flexural, longitudinal, and transverse acoustic modes (ZA, LA, and TA respectively). **d** Variation of the calculated ballistic and intrinsic second sound limits with temperature. **e** Dispersion of second sound at 100 K. **f** Variation of the calculated normalized dip depth with grating period at 100 K, 150 K, and 200 K.

As the grating period decreases, more phonons have MFPs exceeding the period and therefore contribute to ballistic transport (Fig. 3a), hence the wave propagation velocity increases from the bulk second sound velocity to the thermal zero sound velocity (Fig. 2a). For the same reason, the second sound propagation length increases with longer grating period (Fig. 2b), as more phonons participate in the second sound wave type of propagation rather than ballistic transport (Fig. 3a). The velocity of the wavelike mode at fixed grating period monotonically decreases with increasing temperature (Fig. 2a and Fig. S6a), as less phonons participate in ballistic transport at higher temperature due to increasing phonon scattering rates (Fig. 3a).

The second-sound strength, as measured by the normalized dip depth, is greatest at an intermediate grating spacing (Fig. 2f). This can be understood as follows. When the spacing is large, R-scattering dominates. The increased diffusive transport leads to a smaller dip depth. When the spacing is small, ballistic phonons, which do not dip much below zero due to superposition of three different polarizations, mix with second sound waves and diminish the measured normalized dip depth. The thermal wave we observed at 200 K mainly comes from second sound, which is supported by the large difference between the experimental signals and the ballistic limit in Fig. 1d and the MFP analysis in SM 7.

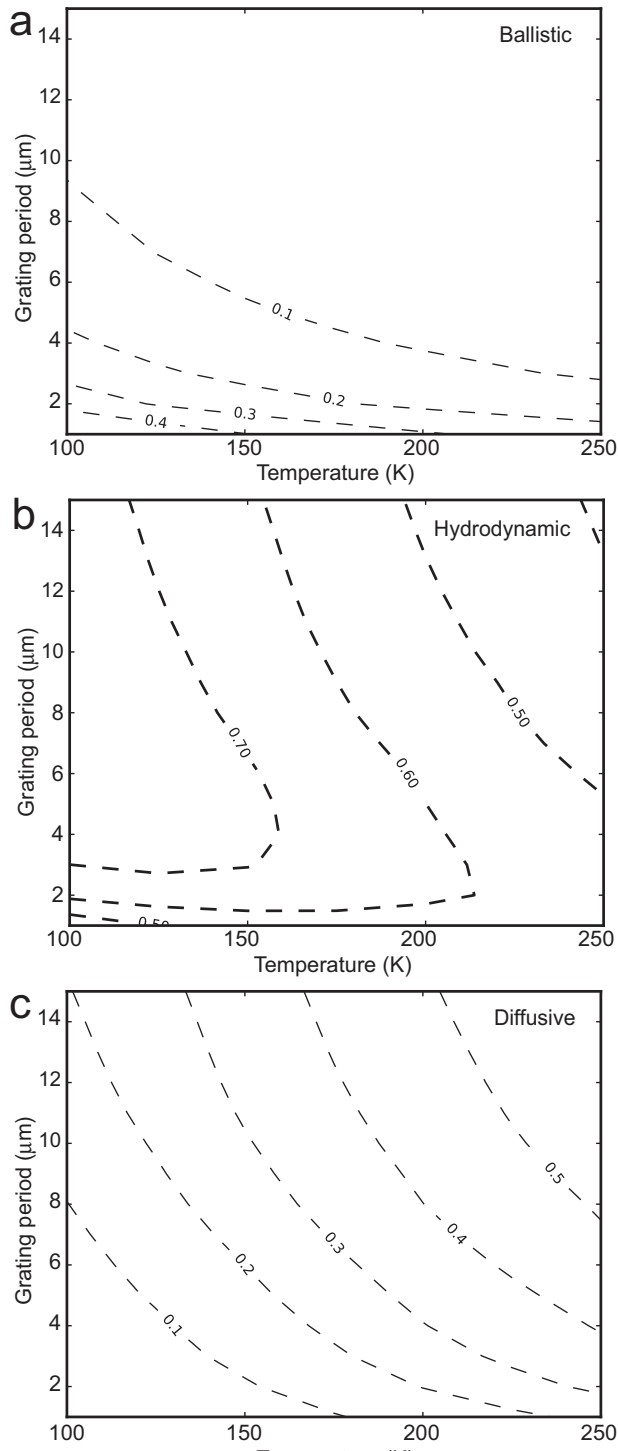

**Fig. 3 The three groups of phonons.** Approximated fraction of excited phonons in the (**a**) ballistic, (**b**) hydrodynamic, and (**c**) diffusive regime based on Eq. (4).

**Strong isotope effect.** Large enhancement in the thermal conductivity via isotope enrichment has been reported in high-thermal-conductivity materials[49]. Isotope scattering is a R-scattering process and can significantly influence second sound. Figure 4a and b compare simulated wave propagation speed and propagation length in isotopically-enriched graphite with natural graphite. In isotopically enriched graphite, the propagation speed is more than 15% higher than in natural graphite across all

grating periods at 125 K, and larger enhancements in the propagation length are observed. Moreover, in isotopically pure samples, our simulations indicate that we could observe second sound even at room temperature at the grating periods of 1.0 $\mu m$ and 1.5 $\mu m$ (Fig. 4c, d). The smaller measured wave propagation speed (Fig. 4a) in natural graphite than in isotopically enriched graphite can be understood as arising from the increased contribution of diffusive phonons.

## Discussion

The picture of three groups of phonons reduces to Fourier diffusion when diffusive phonons dominate. In the TTG experiment, thermal transport can be observed in the form of second sound when hydrodynamic phonons are dominant and thermal zero sound when ballistic phonons are dominant. When the three groups of phonons co-exist, the wave propagation velocity increases with decreasing grating period because (1) the thermal zero sound contributes more at shorter grating period, and (2) the thermal zero sound velocity is higher than the intrinsic second sound velocity. The latter is guaranteed because second sound involves mixing of phonons via normal scattering processes. For example, for a Debye material with sound velocity $v$, the velocity of the thermal zero sound should be around $0.7v$, while that of the intrinsic second sound is $v/\sqrt{3}$. (See SM8 for details.)

We directly observe second sound in graphite above 200 K through TTG experiments. The measured TTG signal also includes contributions from thermal zero sound, i.e., the propagation of ballistic phonon thermal waves. The experimental results can be explained by the coexistence of ballistic, diffusive, and hydrodynamic phonons. The measured wave propagation velocity increases with decreasing grating period due to the increased influence of the thermal zero sound. The experimental results in thermal wave dispersion and strength are in qualitative agreement with first-principles simulations. In isotopically pure graphite, we predict that second sound can be observed even at room temperature, and with higher propagation speed and longer propagation length than seen in our measurements to date. A sign reversal in the thermal response due to nonlocal thermal conductivity is also reported in a recent theoretical study[50], however, the nontrivial relation between nonlocal thermal transport and second sound is beyond the scope of our present study.

## Methods
**Sample.** The natural graphite samples used in this work were obtained from Naturally Graphite©. Based on Atomic force microscopy (AFM) images, the average area of grain is estimated as $382 \pm 270\ \mu m^2$ with the grain size larger than 20 $\mu m$. More details are provided in SM9.

**Thermal transient grating measurements.** To capture the fast dynamics of second sound at micrometer transport lengths, we employ femtosecond laser pump-probe spectroscopy. A schematic illustration of our femtosecond laser thermal transient grating experimental setup is shown in Fig. S1. Ultrafast laser pulses (duration about 290 fs) with 515 nm and 532 nm wavelengths are produced by a second harmonic generator from a 1030 nm amplified laser output and by an optical parametric amplifier, respectively. The repetition rate of the laser pulses is 25 KHz. The 515 nm pulse is used as the pump and the 532 nm pulse is used as the probe. The pump pulse is modulated by an optical chopper at a frequency of around 2 KHz and directed into a delay stage, which can introduce up to 16 ns time delay between the pump and probe pulses. The pump and probe laser beams cross vertically at a phase mask, which diffracts them into horizontally separated +1 and −1 orders which are recombined at the sample after passing through a two-lens imaging system. The crossed pump diffraction orders generate the excitation interference pattern at the sample. The +1-probe diffraction order is taken as a reference beam, and its intensity is reduced by an attenuator made of a 100 nm Au film coated on a 170 $\mu m$-thick glass slide. The −1 diffraction order is taken as the probe beam. An uncoated 170 $\mu m$ thick glass slide mounted on a rotation stage is used to adjust the heterodyne phase shift between the probe and the reference fields. All the beams are focused onto the surface of a natural graphite sample mounted in a cryostat chamber, with pump and probe/reference spot sizes (1/$e^2$ diameters) of 120 $\mu m$ and 105 $\mu m$, respectively. The diffracted probe pulse spatially

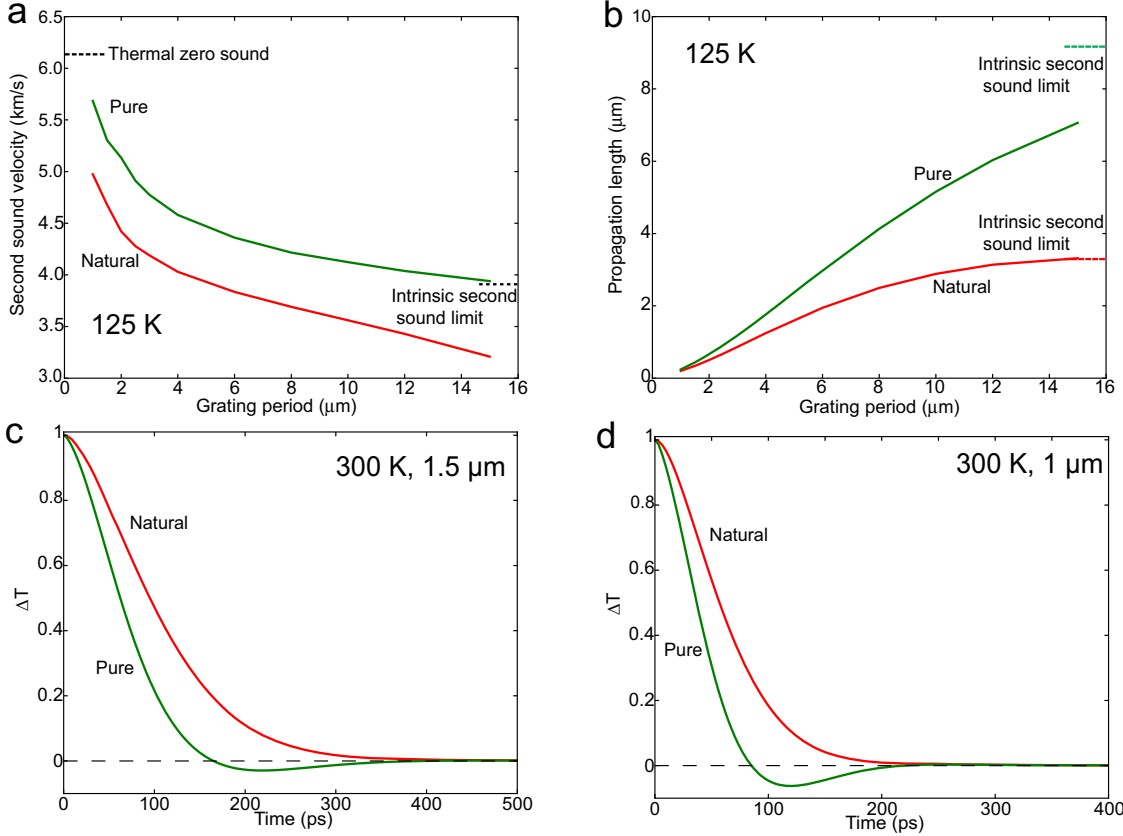

**Fig. 4 Isotope effect on second sound.** Variation of (**a**) second sound velocity and (**b**) propagation length with grating period at 125 K in natural and isotopically pure graphite. **c** Calculated TTG signals at 300 K with a grating period of 1.5 $\mu m$ and (**d**) 1 $\mu m$.

and temporally overlaps with the reflected reference pulse. After passing through Lens 2, the diffracted probe pulse and reflected reference are redirected by a mirror and collected by a photodetector as the heterodyned TTG diffraction signal. The output of the photodetector is analyzed by a lock-in amplifier, which is synchronized with the optical chopper. The pump and probe pulse energies are set at 70 nJ and 52 nJ, respectively. The surface temperature rise due to the pulses is estimated to be <3 K.

**Computational details**. All the first-principle calculations are performed by Vienna Ab Initio Package[51–53] with projector-augmented-wave pseudopotentials and local density approximation for the exchange-correlation energy functional. To include the nonlocal vdW interactions, we use an explicit nonlocal density functional named optB88 functional[54,55]. The geometrical optimization of the unit cell was performed with a $24 \times 24 \times 10$ grid of k-point sampling. The second-order (third-order) force constants were calculated using a real space supercell approach with a $5 \times 5 \times 2$ ($4 \times 4 \times 2$) supercell and $6 \times 6 \times 6$ ($8 \times 8 \times 6$) k-grid. The Phonopy package[56] was used to obtain the second-order force constants. The thirdorder.py and ShengBTE packages[57] were used to obtain the third-order force constants and the phonon scattering matrix on a $16 \times 16 \times 8$ wavevector mesh.

**Temperature response calculation**. The calculation is based on the temperature response with an arbitrary heating profile derived earlier[45]. An expression for the temperature response function for a general heating profile is given as:

$$\triangle \widetilde{T}(\Omega, \mathbf{k}) = \widetilde{Q}(\Omega, \mathbf{k}) \frac{\text{sum}[\mathbf{A}^{-1}\vec{p}]}{\text{sum}[i\mathbf{A}^{-1}D\vec{c}]} \quad (5)$$

where $\Omega$ and $\mathbf{k}$ are the frequency and wavevector from the Fourier transform. We define the sum operation of a vector to add up the values of its elements, i.e., $\text{sum}[\mathbf{a}] = \sum_\mu a_\mu$. The details of Eq. (5) can be found in SM2.

To obtain the temperature response in a TTG measurement with grating period $l$, we need to substitute the following heating profile [Eq. (6)] into Eq. (5) and take the inverse Fourier transform:

$$\widetilde{Q}(\Omega, \mathbf{k}) = \delta\left(\frac{2\pi}{l}\right) + \delta\left(-\frac{2\pi}{l}\right) \quad (6)$$

**Second sound as damped temperature-wave equation**. As discussed earlier[47], using the energy and momentum conservation equations derived from the PBTE with the Callaway model, a damped temperature-wave equation can be derived as Eq. (7):

$$\frac{\partial^2 T}{\partial t^2} + \frac{1}{\tau_{ss}}\frac{\partial T}{\partial t} - v_{ss}^2 \frac{\partial^2 T}{\partial x^2} = 0 \quad (7)$$

with the second sound velocity $v_{ss}$ and relaxation time $\tau_{ss}$ given by Eq. (8).

$$v_{ss}^2 = \frac{\left\langle \frac{C_q q_x v_x}{\omega_q} \right\rangle^2}{\left\langle \frac{C_q q_x^2}{\omega_q^2} \right\rangle \langle C_q \rangle} \quad (8a)$$

$$\tau_{ss} = \frac{\left\langle \frac{C_q q_x^2}{\omega_q^2} \right\rangle}{\left\langle \frac{C_q q_x^2}{\omega_q^2 \tau_{qR}} \right\rangle} \quad (8b)$$

However, Eq. (8) does not capture the wavevector dependence of the decay length and or the second sound speed. Thus, we refer to the second sound velocity obtained from Eq. (8) as the intrinsic limit.

**Ballistic limit**. In the ballistic regime, phonons move without interactions. Each phonon at a given wavevector **q** will simply move along the x-axis with a velocity equal to the x-component of the group velocity $v_x$. If we assume the initial heating is thermally distributed, i.e., each mode is excited proportional to its contribution to heat capacity, then the temperature response of the TTG can be expressed as:

$$\triangle T = 2Q \frac{\left\langle C_q \cos(q v_x t) \right\rangle}{C^2} \quad (9)$$

where $Q$ is the total heating and $C$ is total heat capacity. The TTG temperature response function at the ballistic limit at different temperatures obtained with Eq. (9) is shown in Fig. 2c. The time of the first dip $t_d$ can be obtained by setting the derivative of $\triangle T$ to zero [Eq. (10)]:

$$\frac{d\triangle T}{dt}\bigg|_{t=t_d} = -2Q \frac{\left\langle C_q q v_x \sin(q v_x t_d) \right\rangle}{C^2} = 0 \quad (10)$$

Then thermal zero sound velocity can be obtained as Eq. (11):

$$v_{tzs} = \frac{l}{2t_d} \tag{11}$$

The detailed derivation of Eq. (9) is provided in SM3.

**Dispersion of measured thermal wave**. If the second sound wave is approximated as a damped wave, then the TTG signal can be written as Eq. (12):

$$\triangle T(t) = \exp(-\Omega_i t)\cos(\Omega_r t) \tag{12}$$

The Fourier transform is shown in Eq. (13):

$$\widetilde{\triangle T}(\omega) = \frac{1}{\Omega_i + i(\omega - \Omega_r)} + \frac{1}{\Omega_i + i(\omega + \Omega_r)} \tag{13}$$

This is sharply peaked near $\Omega_r$; near this frequency, the frequency spectrum can be approximated as Eq. (14):

$$|\widetilde{\triangle T}(\omega)|^2 \approx \frac{1}{\Omega_i^2 + (\omega - \Omega_r)^2} \tag{14}$$

Therefore, the frequency of the temperature wave could be obtained by Lorentzian fitting of the temperature response function at a specified wavevector given by Eq. (5). The detailed derivation is provided in SM4.

**Reporting summary**. Further information on research design is available in the Nature Research Reporting Summary linked to this article.

## Data availability
The data that support the findings of this study are available from the corresponding author on reasonable request.

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

## Acknowledgements

We thank Qichen Song for the optical image and Kuan Qiao for the AFM measurement. This work is supported in part by the Office of Naval Research under MURI grant N00014-16-1-2436 via UT Austin (G.C.), by National Science Foundation grant EFRI-1542864, and by the John W. Jarve (1978) Seed Fund for Science Innovation (K.A.N.).

## Author contributions

Z.D. and K.C. contributed equally to this work. K.C. and Z.D. carried out the experiment. B.S. and K.C. built the setup. Z.D. performed the first-principles calculations. Z.D. and G.C. analyzed the data and wrote the paper, with contribution from K.C., B.S., J.S., A.A.M., and K.A.N. G.C. and K.A.N. supervised the research. All authors comment on, discussed, and edited the paper.

## Competing interests

The authors declare no competing interests.
