## [Peer review file · Nature Communications]

REVIEWER COMMENTS

Reviewer #1 (Remarks to the Author):

The authors report the observation of second sound in graphite at the record-high temperature of 200 K. In addition, the authors use first-principles calculations to simulate thermal transient grating experiments, finding satisfiable agreement with their measurements (in the range 100 - 250 K) and, most importantly, predicting the emergence of second sound at room temperature in isotopically pure graphite. An effort is performed to intuitively explain the emergence of second sound from the interplay among three fluids formed by ballistic, diffusive, and hydrodynamic phonons, respectively.

The work is original. The measurements reported are of great significance to the field since they establish a new record for the highest temperature at which second sound is observed. The experimental methodology employed is sound and improves the approach used in the recent, pioneering work [Huberman, S. et al. *Science* (80). 364, 375–379 (2019)]. The experimental results are adequately supported by state-of-the-art first-principles calculations, which simulate thermal transient grating experiments using a Green's function solution of the linearized BTE with the full scattering matrix.

If possible, further details should be provided in the methods to ensure the reproducibility of the work. In particular:

1) In the section "Sample" (line 277), it would be desirable to report more information about the sample used. If possible, it would be useful to know the size of the grains, since a recent work suggests that the emergence of heat hydrodynamics in graphite requires samples with grains' size equal or larger than the ten-micrometer length scale [Jeong et al., *Phys. Rev. Lett.* 127, 085901 (2021)] and the former, pioneering work of the authors [Huberman, S. et al. *Science* (80). 364, 375–379 (2019)] reported the emergence of second sound in a graphite sample having a typical grain size equal to 10 micrometers.

2) In the section "Computational details" (line 306) it would be desirable to add more details on the parameters used in the first-principles calculations, to ease the reproducibility of the theoretical analysis.

Overall, I believe this work is of great significance and I strongly recommend its publication in *Nature Communication* (after the minor revision suggested above).

In addition, I have the following additional suggestions for minor changes:

a) I would suggest specifying that the second sound measured here is of "drifting" type [Hardy, *Phys. Rev. B* 2, 1193 (1970)], to distinguish it from the "driftless" second sound measured in a rapidly varying temperature field [Beardo et al., *Science Advances* 7, 2021].

b) I would suggest removing the word "acoustic" at line 26, since in general also optical phonons can contribute to thermal transport.

c) At line 30 there is a typo (the correct sentence should probably read "leading to an effective thermal conductivity value that diminishes as the length scale...").

d) At line 55 and 56, when discussing the expectations about the emergence of second sound at temperatures higher than 100 K for TTG spatial periods smaller than those used in their previous study, the authors should specify the origin of these expectations. I guess that their expectations were based on the first-principles calculations reported in Fig 4 of their previous work [Huberman, S. et al. *Science* (80). 364, 375–379 (2019)]. It might be worth mentioning that the expectation of requiring

smaller length scales to observe hydrodynamic behavior at higher temperatures in graphite is supported also by another recent work [Simoncelli, M. et al. Phys. Rev. X. 10, 011019 (2020)] (see in particular Fig. 5 of this reference).

e) At line 174, there is a typo (a period).

f) After equation S1, almost all the quantities appearing in such an equation are specified, with the exception of "N" and "v". Specifying that "N" is the number of discretized points in the Brillouin zone and "v" is the unit cell volume, would allow the reader to avoid looking for the significance of these parameters in Ref. 4 [Chiloyan, V. et al. arXiv Prepr. arXiv1711.07151 (2017)].

g) In the title of panel b) of Fig. S6 there is a typo ("Popogation").

Reviewer #2 (Remarks to the Author):

This manuscript is an extension of the group's previous work published in Science (Ref. 21) - observation of the 2nd sound in 2D material - graphite. I don't think the current manuscript deserves publication in Nature Communication due to the following reasons:

(1) It doesn't provide any new insights/science into the problem. It just extends the temperature of observation from 100 K (Science 2019) to 200 K which I think is not really exciting. In the current work, also in previous work (ref. 21, Science, 2019), the authors attribute the oscillatory decay (with negative T change) of the transient grating signal as a smoking-gun evidence of second sound propagation. However, in a recent theoretical work (Phys. Rev. B 102, 104310), it is shown that similar oscillations are observed in diamond and silicon at 100 and 50 K, respectively. It seems out of the temperature range of second sound propagation in these two materials. Moreover, the propagation speed of the signal seems to be much smaller than that of second sound. My main question is: is second sound the only possible explanation of the oscillatory response (with negative T change)? Is there any other possible mechanism that could in principle generate similar response? The author may consider elaborate on this point further in the manuscript.

(2) The method and technique used are also quite similar to the previous one published in Science by the same group. The current manuscript has improved the transient grating period down to 2 micron. Their work reported in Science (ref. 21) was 6 micron. With this progress, they are further able to tune the participation ratio of ballistic, hydrodynamic and diffusive phonons, and study the resultant heat transport. This technique increment does not provide further/deeper science. It just provides similar results as in Ref. 21. The results maybe more suited for a specialized journal

Other specific questions are:

(3) The expression for the second sound in Eq. (8) looks a bit different from that in Ref. 13. Are they the same?

(4) A minor point on the terminology the authors may consider (optional): Normally, the word 'fluid' is used for system with strong inter-quasiparticle interaction. Here, is it appropriate to take the ballistic phonons as one type of fluid?

(5) It would be beneficial for the readers if the authors could give more details on the DFT calculation, i.e. the sampling mesh used, etc..

(6) Citations of some references such as Ref.37, 46 are not consistent with others.

Reviewer #3 (Remarks to the Author):

This manuscript is an extension of the group's previous work published in Science (Ref. 21) - observation of the 2nd sound in 2D material – graphite. I don't think the current manuscript deserves publication in Nature Communication due to the following reasons:

- (1) It doesn't provide any new insights/science into the problem. It just extends the temperature of observation from 100 K (Science 2019) to 200 K which I think is not really exciting. In the current work, also in previous work (ref. 21, Science, 2019), the authors attribute the oscillatory decay (with negative T change) of the transient grating signal as a smoking-gun evidence of second sound propagation. However, in a recent theoretical work (Phys. Rev. B 102, 104310), it is shown that similar oscillations are observed in diamond and silicon at 100 and 50 K, respectively. It seems out of the temperature range of second sound propagation in these two materials. Moreover, the propagation speed of the signal seems to be much smaller than that of second sound. My main question is: is second sound the only possible explanation of the oscillatory response (with negative T change)? Is there any other possible mechanism that could in principle generate similar response? The author may consider elaborate on this point further in the manuscript.
- (2) The method and technique used are also quite similar to the previous one published in Science by the same group. The current manuscript has improved the transient grating period down to 2 micron. Their work reported in Science (ref. 21) was 6 micron. With this progress, they are further able to tune the participation ratio of ballistic, hydrodynamic and diffusive phonons, and study the resultant heat transport. This technique increment does not provide further/deeper science. It just provides similar results as in Ref. 21. The results maybe more suited for a specialized journal

Other specific questions are:

- (3) The expression for the second sound in Eq. (8) looks a bit different from that in Ref. 13. Are they the same?
- (4) A minor point on the terminology the authors may consider (optional): Normally, the word 'fluid' is used for system with strong inter-quasiparticle interaction. Here, is it appropriate to take the ballistic phonons as one type of fluid?
- (5) It would be beneficial for the readers if the authors could give more details on the DFT calculation, i.e. the sampling mesh used, etc..
- (6) Citations of some references such as Ref.37, 46 are not consistent with others.

The authors observed second sound in graphite at temperatures of over 200 K using a sub-picosecond transient grating technique. The experimentally determined dispersion relations of the thermal-wave velocity increases with the decreasing of grating period, consistent with first-principles-based solution of the Peierls-Boltzmann transport equation. Through simulation, they reveal this increase resulting from thermal zero sound—the thermal waves due to ballistic phonons. Their experimental findings are explained with the interplay among “three fluids”: ballistic, diffusive, and hydrodynamic phonons. Their ab initio calculations further predict a large isotope effect on the properties of thermal waves and the existence of second sound at room temperature in isotopically pure graphite. The experimental discovery of the second sound in graphite at "high temperature" merits its publication in Nature Communications if the following points are considered in revising their manuscript.

1. There is a recent report of observation of second sound in graphite ribbon by heat-pulse experiment (J. Jeong *et al.* Phys. Rev. Lett. **127**, 085901, 2021). There is also a recent theoretical report of direct simulation of second sound in graphene ribbon (X.-P. Luo *et al.* Phys. Rev. B. **100**, 155401, 2019), including the case of TTG set-up. The authors shall mention those works and have some discussions in the manuscript.
2. The authors observe a second sound propagation in graphite with natural abundance even around 200K. The detailed information about the MFP of isotope scattering shall be provided to show that it is indeed much larger than the grating period length. Or the authors shall show that it is much larger than the MFP of normal scattering. Otherwise, the hydrodynamic transport might be destroyed by the isotope scattering as shown in previous theoretical studies.
3. The reviewer cannot agree with the terminology of ‘three fluids’ (ballistic, hydrodynamic, diffusive phonons). There is only one kind of quantized excitation here, i.e. phonons, which may transport in different ways, as is well known already. The classical two-fluid model proposed by Landau for the study of second sound in liquid helium refer to two components: super-fluid and normal-fluid. If ‘three fluids’ is adopted here, the widely studied quasi-ballistic (or ballistic-diffusive) heat transport would be explained by a ‘two-fluid’ model, which seems a bit confusing or misleading.
4. Why is the ballistic limit (Fig. 2c) calculated with a grating period of $20\mu\text{m}$? Physically speaking, it shall be in the diffusive regime at 100K. Could a realistic ballistic limit be calculated, for instance, with extremely small grating period?
5. The authors show discrete points of experimental results. How large is the errorbar of those data? It is important to make sure the amplitude of second sound oscillation is larger than the errorbar.

6. Minor point: the reviewer suggests the authors to carefully check the reference to the subfigures in the main text. For instance, in Line 136 on Page 5, 'Fig. 1f' shall be 'Fig. 1d' instead.

Revision report for NCOMMS-21-33351

Observation of Second Sound in Graphite over 200 K

Zhiwei Ding,^{1*} Ke Chen,^{1*} Bai Song,¹ Jungwoo Shin,¹ Alexei A. Maznev,² Keith A. Nelson,^{2†} and Gang Chen^{1‡}

¹Department of Mechanical Engineering

²Department of Chemistry

Massachusetts Institute of Technology

Cambridge, MA 02139

We greatly appreciate the reviewers for their critical comments, which helped us to improve our manuscript. We have addressed the reviewers' comments below point-by-point, and revised the manuscript accordingly. For clarity, *the reviewers' comments are included in italic*, while **our responses are in regular font and blue**. **Modifications to the manuscript are in red**. A copy of the manuscript with the revisions highlighted in **yellow** is also provided for reference.

* These authors contributed equally

† Corresponding author: kanelson@mit.edu

‡ Corresponding author: gchen2@mit.edu

Reviewer: 1

The authors report the observation of second sound in graphite at the record-high temperature of 200 K. In addition, the authors use first-principles calculations to simulate thermal transient grating experiments, finding satisfiable agreement with their measurements (in the range 100 - 250 K) and, most importantly, predicting the emergence of second sound at room temperature in isotopically pure graphite. An effort is performed to intuitively explain the emergence of second sound from the interplay among three fluids formed by ballistic, diffusive, and hydrodynamic phonons, respectively.

The work is original. The measurements reported are of great significance to the field since they establish a new record for the highest temperature at which second sound is observed. The experimental methodology employed is sound and improves the approach used in the recent, pioneering work [Huberman, S. et al. Science (80). 364, 375–379 (2019)]. The experimental results are adequately supported by state-of-the-art first-principles calculations, which simulate thermal transient grating experiments using a Green's function solution of the linearized BTE with the full scattering matrix.

Overall, I believe this work is of great significance and I strongly recommend its publication in Nature Communication (after the minor revision suggested above).

Our response:

We greatly appreciate the reviewer for the encouraging comments, for helping us improve our manuscript, and for the strong recommendation for publication.

If possible, further details should be provided in the methods to ensure the reproducibility of the work. In particular:

1) In the section "Sample" (line 277), it would be desirable to report more information about the sample used. If possible, it would be useful to know the size of the grains, since a recent work suggests that the emergence of heat hydrodynamics in graphite requires samples with grains' size equal or larger than the ten-micrometer length scale [Jeong et al., Phys. Rev. Lett. 127, 085901 (2021)] and the former, pioneering work of the authors [Huberman, S. et al. Science (80). 364, 375–379 (2019)] reported the emergence of second sound in a graphite sample having a typical grain size equal to 10 micrometers.

Our response:

We agree with the reviewer's comment that the grain size of graphite plays an important role in the experimental detection of hydrodynamics. We observed several graphite crystallites larger than 1 mm in length in the optical images as shown in Fig. S11a. We performed atomic force microscopy (AFM) to further analyze the surface defects, step edges and grain boundaries Fig. S11b. We used ImageJ to estimate the average size of grains that are surrounded by the step edge or grain boundaries in the AFM images. The average area of grain is estimated as $382 \pm 270 \mu\text{m}^2$ and the grain size is in general larger than 20 μm .

We added the following to the revised manuscript. In the Sample section, more details on the sample are provided:

Based on Atomic force microscopy (AFM) images, the average area of grain is estimated as $382 \pm 270 \mu\text{m}^2$ with the grain size larger than $20 \mu\text{m}$. More details are provided in SM9.

A new session is added in SM to discuss the characterization of the sample.

9. Characterization of the sample

We observe several graphite crystallites larger than 1 mm in length, which are highlighted in different colors in Fig. S11(a). We performed atomic force microscopy (AFM) to further analyze the surface defects, step edges and grain boundaries. Fig. S11(b) shows AFM height image of the natural graphite crystal. The dashed arrows show step edges where the height changes across the lines. The solid arrows indicate grain boundaries where height does not vary across the line. ImageJ was used to estimate the average area of grains that are surrounded by the step edge or grain boundaries in the AFM image. The average grain area is estimated as $382 \pm 270 \mu\text{m}^2$ where the largest longer axis could be larger than $100 \mu\text{m}$ and the typical grain size is larger than $20 \mu\text{m}$.

Figure S11 (a) Optical image of the natural graphite crystal. Individual graphite flakes are highlighted in different colors. (b) AFM z-height image of the natural graphite crystal. The dashed arrows show step edges where the height changes across the lines. The solid arrows indicate grain boundaries where height does not vary across the line.

2) In the section “Computational details” (line 306) it would be desirable to add more details on the parameters used in the first-principles calculations, to ease the reproducibility of the theoretical analysis.

Our response:

We thank the reviewer for the constructive suggestion. The following computational details have been included in the Computational details section.

Details pertaining to the relaxed structure and force constants:

The geometrical optimization of the unit cell was performed with a $24 \times 24 \times 10$ grid of k-point sampling. The second-order (third-order) force constants were calculated using a real space supercell approach with a $5 \times 5 \times 2$ ($4 \times 4 \times 2$) supercell and $6 \times 6 \times 6$ ($8 \times 8 \times 6$) k-grid.

Details on the scattering matrix:

a $16 \times 16 \times 8$ wavevector mesh

In addition, I have the following additional suggestions for minor changes:

a) I would suggest specifying that the second sound measured here is of “drifting” type [Hardy, Phys. Rev. B 2, 1193 (1970)], to distinguish it from the “driftless” second sound measured in a rapidly varying temperature field [Beardo et al., Science Advances 7, 2021].

Our response:

We thank the reviewer for his/her constructive suggestions. We have added the following statement to Page 4:

What is reported here is the “drifting” second sound as discussed by Hardy⁵ where collective drift motion of phonons due to strong N-scattering is essential. This “drifting” second sound is distinguished from “driftless” second sound suggested by a recent report on measurements in a rapidly varying temperature field⁴².

b) I would suggest removing the word “acoustic” at line 26, since in general also optical phonons can contribute to thermal transport.

Our response:

We thank the reviewer for the suggestions. We have deleted the word “acoustic”.

c) At line 30 there is a typo (the correct sentence should probably read “leading to an effective thermal conductivity value that diminishes as the length scale...”).

Our response:

We thank the reviewer for his/her careful checking. We have corrected the typo by removing the “s” from “values” as suggested by the reviewer and changed “diminish” to “diminishes”.

d) At line 55 and 56, when discussing the expectations about the emergence of second sound at temperatures higher than 100 K for TTG spatial periods smaller than those used in their previous study, the authors should specify the origin of these expectations. I guess that their expectations were based on the first-principles calculations reported in Fig 4 of their previous work [Huberman, S. et al. Science (80). 364, 375–379 (2019)]. It might be worth mentioning that

the expectation of requiring smaller length scales to observe hydrodynamic behavior at higher temperatures in graphite is supported also by another recent work [Simoncelli, M. et al. Phys. Rev. X. 10, 011019 (2020)] (see in particular Fig. 5 of this reference).

Our response:

We thank the reviewer for the constructive suggestion. We have added the following words and references to Page 3.

as suggested by the theoretical calculations^{21,32}

e) At line 174, there is a typo (a period).

Our response:

We thank the reviewer. We have corrected the typo

f) After equation S1, almost all the quantities appearing in such an equation are specified, with the exception of “N” and “v”. Specifying that “N” is the number of discretized points in the Brillouin zone and “v” is the unit cell volume, would allow the reader to avoid looking for the significance of these parameters in Ref. 4 [Chiloyan, V. et al. arXiv Prepr. arXiv1711.07151 (2017)].

Our response:

We thank the reviewer for his/her time. We have added the definition of N and v to line 47 of the supplementary material:

N is the number of discretized points in the Brillouin zone, v is the unit cell volume

g) In the title of panel b) of Fig. S6 there is a typo (“Popogation”).

Our response:

We thank the reviewer for his/her careful reading. We have corrected Fig. S6 as follows:

Figure S6 Calculated dependence of (a) second sound speed and (b) propagation length on the grating period at different temperatures.

Reviewer #2 (Remarks to the Author):

This manuscript is an extension of the group's previous work published in Science (Ref. 21) - observation of the 2nd sound in 2D material – graphite. I don't think the current manuscript deserves publication in Nature Communication due to the following reasons:

Our response:

We thank the reviewer for his/her time spent reviewing this work. However, we respectfully disagree with the comments, as explained by our responses in the following.

(1) It doesn't provide any new insights/science into the problem. It just extends the temperature of observation from 100 K (Science 2019) to 200 K which I think is not really exciting. In the current work, also in previous work (ref. 21, Science, 2019), the authors attribute the oscillatory decay (with negative T change) of the transient grating signal as a smoking-gun evidence of second sound propagation. However, in a recent theoretical work (Phys. Rev. B 102, 104310), it is shown that similar oscillations are observed in diamond and silicon at 100 and 50 K, respectively. It seems out of the temperature range of second sound propagation in these two materials. Moreover, the propagation speed of the signal seems to be much smaller than that of second sound. My main question is: is second sound the only possible explanation of the oscillatory response (with negative T change)? Is there any other possible mechanism that could in principle generate similar response? The author may consider elaborate on this point further in the manuscript.

Our response:

We thank the reviewer for his/her critical comments. We believe the reviewer's comments involve two major aspects and would like to address them one by one as follows:

“It doesn't provide any new insights/science into the problem.”

(1) In this work, we employed sub-picosecond transient grating technique to enable direct observation of second sound at a record-high temperature of over 200 K. The prospects for reaching room temperature, at which revolutionary new strategies for thermal management would become possible, suddenly loom large. With the enlarged grating-period window, we reported the dispersion of the thermal wave, whose velocity increases with decreasing grating period due to the increasing contribution of thermal zero sound—the thermal wave due to ballistic phonons. In addition, we predict the possible observation of second sound at room temperature in isotopically pure graphite.

“My main question is: is second sound the only possible explanation of the oscillatory response (with negative T change)? Is there any other possible mechanism that could in principle generate similar response?”

(2) Thermal zero sound can also cause oscillations, however, whether the thermal response can dip below zero or not depends on the details of the phonon spectrum. Previous modeling¹ showed that under a constant-phonon-group-velocity approximation, the transient grating

response in the ballistic limit can dip below zero, but this is no longer the case if phonon dispersion is taken into account. Our calculations in Fig. 1d show that in graphite, the ballistic signals cannot account for the observed dip in the thermal response. Although we cannot rule out the existence of materials that can show thermal zero sound leading to a negative dip in the thermal response, we believe that our experiments in conjunction with our simulations confirm that the negative dip in our case is indeed due to second sound. As for the oscillatory response in diamond and silicon reported in the theoretical work Phys. Rev. B 102, 104310, the authors attribute it to non-local thermal conductivity. They are not “drifting” second sound signals because the normal-scattering rates do not dominate over momentum-destroying scattering rates in these materials. As a result, there is no contradiction with our current work. We agree with this PRB paper that “the relation between the nonlocal lattice thermal conductivity and hydrodynamic phonon transport is nontrivial”, and have added the following sentence to the revised manuscript.

A sign reversal in the thermal response due to non-local thermal conductivity is also reported in a recent theoretical study⁴⁶, however, the nontrivial relation between non-local thermal transport and second sound is beyond the scope of our present study.

(2) The method and technique used are also quite similar to the previous one published in Science by the same group. The current manuscript has improved the transient grating period down to 2 micron. Their work reported in Science (ref. 21) was 6 micron. With this progress, they are further able to tune the participation ratio of ballistic, hydrodynamic and diffusive phonons, and study the resultant heat transport. This technique increment does not provide further/deeper science. It just provides similar results as in Ref. 21. The results maybe more suited for a specialized journal

Our response:

We respectfully disagree with the reviewer’s concern about the novelty, and would like to highlight two points here.

First, our experimental technique is nontrivial. Conceptually speaking, it is true that both the current manuscript and our previous work (Ref. 21) employed the method of transient thermal grating (TTG). However, in practice, the two platforms are substantially different. Our previous platform employed a continuous wave (CW) probe, with a fast photodiode and a fast oscilloscope. This setup was relatively straightforward but had a temporal resolution of ~ 0.5 ns, which only allowed us to use the grating periods as small as 6 μm . In order to detect temperature waves at smaller length scales, we replaced the CW probe with a femtosecond pulsed probe as explained in the Materials and methods section. For detection, a pulsed probe further required an optical delay stage in conjunction with a lock-in amplifier, and mechanical chopper to synchronize the pump pulse. With this new approach, we were finally able to go down to a grating period of about 2 μm . None of this is as trivial as it might seem to be.

Further, our experimental results and the corresponding analyses are not similar to Ref. 21. The reviewer’s concern seems to have arisen from the fact that our measured data agree well with our previous theoretical modeling. However, this agreement should instead speak to the strength of

our work. Any theoretical modeling, as plausible as it may appear, must be validated by rigorous experiments before it is accepted. In fact, to continue along this line of research, we believe it remains of great importance to probe thermal transport at even smaller length scales, from sub-micrometer to nanometer. Of course, this requires further innovations on the experimental side.

In addition, the reported result here is not trivial as acknowledged by the reviewer: "With this progress, they are further able to tune the participation ratio of ballistic, hydrodynamic and diffusive phonons, and study the resultant heat transport." It is the great progress in the measurement technique that has led to our important physical observations. We believe that the observation of dispersion in second sound velocity is not a simple technical work but an important footprint of thermal science. As stated by the third reviewer, "*The experimental discovery of the second sound in graphite at "high temperature" merits its publication in Nature Communications.*".

Other specific questions are:

(3) The expression for the second sound in Eq. (8) looks a bit different from that in Ref. 13. Are they the same?

Our response:

The expression for second sound velocity provided by Eq. (8a), is indeed identical to Eq. (20) of Ref. 13.

(4) A minor point on the terminology the authors may consider (optional): Normally, the word 'fluid' is used for system with strong inter-quasiparticle interaction. Here, is it appropriate to take the ballistic phonons as one type of fluid?

Our response:

We thank the reviewer for his/her constructive suggestions. We revised the manuscript, we have changed the term 'fluid' with "three groups of phonons" to avoid possible confusion.

(5) It would be beneficial for the readers if the authors could give more details on the DFT calculation, i.e. the sampling mesh used, etc..

Our response:

We thank the reviewer for the constructive suggestion. The following computational details have been included in the computational detail session.

Details pertaining to the relaxed structure and force constants:

“The geometry optimization of the unit cell was performed with a $24 \times 24 \times 10$ grid of k-point sampling. The second-order (third-order) force constants were calculated using a real space supercell approach with a $5 \times 5 \times 2$ ($4 \times 4 \times 2$) supercell and $6 \times 6 \times 6$ ($8 \times 8 \times 6$) k-grid.”

Details on the scattering matrix:

“a $16 \times 16 \times 8$ wavevector mesh”

(6) Citations of some references such as Ref.37, 46 are not consistent with others.

Our response:

We thank the reviewer for the careful checking. In the revised manuscript, we ensured a uniform format of the references.

Reviewer #3 (Remarks to the Author):

The authors observed second sound in graphite at temperatures of over 200 K using a sub-picosecond transient grating technique. The experimentally determined dispersion relations of the thermal-wave velocity increases with the decreasing of grating period, consistent with first-principles-based solution of the Peierls-Boltzmann transport equation. Through simulation, they reveal this increase resulting from thermal zero sound—the thermal waves due to ballistic phonons. Their experimental findings are explained with the interplay among “three fluids”: ballistic, diffusive, and hydrodynamic phonons. Their ab initio calculations further predict a large isotope effect on the properties of thermal waves and the existence of second sound at room temperature in isotopically pure graphite. The experimental discovery of the second sound in graphite at “high temperature” merits its publication in Nature Communications if the following points are considered in revising their manuscript.

Our response:

We thank the reviewer for his/her very positive evaluation and constructive suggestions.

1. There is a recent report of observation of second sound in graphite ribbon by heat-pulse experiment (J. Jeong et al. Phys. Rev. Lett. 127, 085901, 2021). There is also a recent theoretical report of direct simulation of second sound in graphene ribbon (X.-P. Luo et al. Phys. Rev. B. 100, 155401, 2019), including the case of TTG set-up. The authors shall mention those works and have some discussions in the manuscript.

Our response:

We thank the reviewer for his/her constructive suggestions. In the revised manuscript, the theoretical report of the direct simulation of second sound in graphene ribbon (X.-P. Luo et al. Phys. Rev. B. 100, 155401, 2019) is added in the discussion of the theoretical prediction of second sound as ref. 25 and the recent heat-pulse experiment (J. Jeong et al. Phys. Rev. Lett. 127, 085901, 2021) is mentioned in the overview of experimental studies of second sound as ref. 31. More specifically, we added the theoretical simulation of second sound and the discussion of the recent heat-pulse experiment to Page 2.

However, phonon hydrodynamics at substantially higher temperatures in low-dimensional and van der Waals materials has recently been theoretically predicted and experimentally observed, stimulating renewed interest, especially in second sound^{12,13,19-25}

The observation of second sound in graphite is further confirmed via the transient hydrodynamic lattice cooling by picosecond laser irradiation of graphite³¹

2. The authors observe a second sound propagation in graphite with natural abundance even around 200K. The detailed information about the MFP of isotope scattering shall be provided to show that it is indeed much larger than the grating period length. Or the authors shall show that it is much larger than the MFP of normal scattering. Otherwise, the hydrodynamic transport might be destroyed by the isotope scattering as shown in previous theoretical studies.

Our response:

We thank the reviewer for his/her constructive suggestions. Since the ZA phonon dominates the contribution to second sound, in this revised version, we provide a comparison of the grating period and ZA phonon MFP of different scatterings at 200 K (Fig. S10). A characteristic frequency, $k_B T / 2\pi\hbar$, is also marked, below which phonons can be readily activated at a given temperature. The average phonon MFP is computed as: $\Lambda_l = \langle C_q v \tau_l \rangle / \langle C_q \rangle$. At 200 K, the average N-scattering and R-scattering MFP are around 0.2 μm and 5.6 μm respectively. Based on these details, we demonstrate that grating periods we investigated are indeed larger than the MFP of normal scattering and smaller than the MFP of resistive scattering (including Umklapp scattering and isotopic scattering). We add the following figure and paragraph to Page 9:

7. Mean free path analysis at 200 K

A detailed mean free path analysis is performed to further confirm the observation at 200 K is indeed second sound instead of ballistic signals. Since the ZA phonon dominates the contribution to second sound, we provide a comparison of the grating period and ZA phonon MFP of different scatterings at 200 K (Fig. S10). A characteristic frequency, $k_B T / 2\pi\hbar$, is also marked, below which phonons can be readily activated at a given temperature. The average phonon MFP is computed as: $\Lambda_l = \langle C_q v \tau_l \rangle / \langle C_q \rangle$. At 200 K, the Average N-scattering and R-scattering MFP are around 0.2 μm and 5.6 μm respectively. Therefore, our investigated grating periods i.e. 2~3 μm are indeed larger than the MFP of normal scattering and smaller than the MFP of resistive scattering (including Umklapp scattering and isotopic scattering).

Figure S10. Comparison of mean free path at 200 K (a) N-scattering and I-scattering (b) N-scattering and R-scattering where the characteristic frequency $f_T = kT/2\pi\hbar$ is marked

3. The reviewer cannot agree with the terminology of ‘three fluids’ (ballistic, hydrodynamic, diffusive phonons). There is only one kind of quantized excitation here, i.e. phonons, which may

transport in different ways, as is well known already. The classical two-fluid model proposed by Landau for the study of second sound in liquid helium refer to two components: super-fluid and normal-fluid. If 'three fluids' is adopted here, the widely studied quasi-ballistic (or ballistic-diffusive) heat transport would be explained by a 'two-fluid' model, which seems a bit confusing or misleading.

Our response:

We thank the reviewer for his/her constructive suggestions. In the revised manuscript, we have replaced the term “three fluids” by “three groups of phonons” to avoid possible confusion.

4. *Why is the ballistic limit (Fig. 2c) calculated with a grating period of $20\mu\text{m}$? Physically speaking, it shall be in the diffusive regime at 100K. Could a realistic ballistic limit be calculated, for instance, with extremely small grating period?*

Our response:

We thank the reviewer for his/her suggestions. We agree with the review that the system shall be in the diffusive regime at a grating period of $20\mu\text{m}$. In the revised manuscript, we have changed the grating period to 200 nm. However, in the ballistic limit, the change of the grating period only transforms the time scale, and the shape of the curve remains unchanged. The revised Fig. 2c is shown below:

Figure 2. Dispersion of thermal waves. The measured and calculated data are represented by points and curves, respectively. (a) Dependence of wave velocity on grating period. Also shown are the intrinsic second sound limit from Eq. (2) and the thermal zero sound limit discussed in Methods. Damping reduces the second sound velocity below the intrinsic limit at large grating periods, while zero sound increases the measured wave velocity at small grating periods. (b) Propagation length variation with grating period at different temperatures. (c) Calculated ballistic limit of the TTG signal at 100 K and its contributions from flexural, longitudinal, and transverse acoustic modes (ZA, LA, and TA respectively). (d) Variation of the calculated ballistic and intrinsic second sound limits with temperature. (e) Dispersion of second sound at 100 K. (f) Variation of the calculated normalized dip depth with grating period at 100 K, 150 K and 200 K.”

5. The authors show discrete points of experimental results. How large is the errorbar of those data? It is important to make sure the amplitude of second sound oscillation is larger than the errorbar.

Our response:

We thank the reviewer for his/her good suggestion of providing the information of our measurement error. The statistical error of the data points in the signal waveforms can be estimated by the standard deviation of the negative-time-delay signals, as shown in Fig. S2b. In our pump-probe experiments, the negative-time-delay signals should be 0 in the ideal case. While in the real case, the randomness of the measurement system cause fluctuation around the 0 baseline (Fig. S2b). The magnitude of the fluctuation could be used to estimate the measurement errors. Most of our measurement errors are around 0.01, while the “dip” amplitude of those signals which we claimed to be second sound observations are much larger than their errors, indicating the “dips” are intrinsic and reliable. In the revised manuscript, the following paragraph and updated figure are added to the SM1.

The statistical error of the data points in the signal waveforms can be estimated by the standard deviation of the negative-time-delay signals, as shown in Fig. S2b. In our pump-probe experiments, the negative-time-delay signals should be 0 in the ideal case. While in the real case, the randomness of the measurement system cause fluctuation around the 0 baseline (Fig. S2b). The magnitude of the fluctuation could be used to estimate the measurement errors. Most of our measurement errors are around 0.01, while the “dip” amplitude of those signals which we claimed to be second sound observations are much larger than their errors, indicating the “dips” are intrinsic and reliable.

Figure S2. (a) Measured TTG signals at 200 K and 2um grating period at two heterodyne phases. One phase is set to maximize the second peak, corresponding to the phase grating signal, while the other phase is set to minimize the second peak, corresponding to the amplitude grating signal,

respectively. (b) Normalized phase grating signal and the smoothed response obtained using the Savitzky-Golay method. The negative-time-delay signals are used to estimate the measurement error.

6. *Minor point: the reviewer suggests the authors to carefully check the reference to the subfigures in the main text. For instance, in Line 136 on Page 5, 'Fig. 1f' shall be 'Fig. 1d' instead.*

Our response:

We really appreciate the reviewer for his/her careful reading. We have corrected the typo.

A larger discrepancy between experiment and simulation is observed at 100 K (Fig. 1e) than 200 K (Fig. 1d)

References

1. Collins, K. C. *et al.* Non-diffusive relaxation of a transient thermal grating analyzed with the Boltzmann transport equation. *J. Appl. Phys* **114**, 104302 (2013).

REVIEWERS' COMMENTS

Reviewer #1 (Remarks to the Author):

The authors have provided detailed replies and rebuttals to questions and criticisms raised by each of the referees and made a series of according changes and additions to the manuscript. This has substantially improved the paper.

I recommend the paper for publication without further necessary changes.

Reviewer #2 (Remarks to the Author):

Although the authors have responded all reviewers' critics and comments and revised the manuscript, I still keep my original opinion that I don't recommend the publication of the current manuscript in Nat Comm.

Reviewer #3 (Remarks to the Author):

I think the present version of this manuscript is acceptable for publication in Nature Communications.